# Calcification in Salivary Gland Cancer Mimicking Sialolithiasis—A Diagnostic Pitfall on Imaging: Report of Two Cases and Brief Review of the Literature

**DOI:** 10.3390/jcm11123329

**Published:** 2022-06-10

**Authors:** Vivian Thimsen, Vanessa Fauck, Marco Wiesmüller, Abbas Agaimy, Mirco Schapher, Heinrich Iro, Michael Koch, Konstantinos Mantsopoulos

**Affiliations:** 1Department of Otorhinolaryngology, Head and Neck Surgery, Friedrich-Alexander-Universität Erlangen-Nürnberg (FAU), 91054 Erlangen, Germany; vanessa.fauck@klinikum-nuernberg.de (V.F.); mirco.schapher@klinikum-nuernberg.de (M.S.); heinrich.iro@uk-erlangen.de (H.I.); michael.koch@uk-erlangen.de (M.K.); konstantinos.mantsopoulos@uk-erlangen.de (K.M.); 2Department of Radiology, Friedrich-Alexander-Universität Erlangen-Nürnberg (FAU), 91054 Erlangen, Germany; marco.wiesmueller@uk-erlangen.de; 3Insitut of Pathology, Friedrich-Alexander-Universität Erlangen-Nürnberg (FAU), 91054 Erlangen, Germany; abbas.agaimy@uk-erlangen.de

**Keywords:** sialolithiasis, parotid gland, adenoid cystic carcinoma, acinic cell carcinoma, ultrasound, tumor calcification

## Abstract

***Objectives:*** Sialolithiasis is the most common cause of calcifications detected with ultrasound in patients with chronic inflammatory symptoms and swellings of the salivary glands. Other differential diagnoses of calcifications are extremely rare and mostly benign. ***Methods:*** Case report and literature review. ***Results:*** Two rare cases of malignant parotid gland tumors with calcifications in a localization typical for sialolithiasis, which were mistaken for salivary calculi based on image findings, are presented. ***Conclusions:*** This report intends to highlight the pitfalls in the imaging of parotid gland diseases. Even if malignant tumors of the parotid gland with calcifications are extremely rare, in ambiguous cases, differential diagnoses should be considered carefully. A high suspicion index of the need for further diagnostics in cases with calcifications is practical and could include missing periprandial symptoms, no obstruction signs in the proximal duct, and missing evidence of sialolithiasis in sialendoscopy.

## 1. Introduction

Although uncommon, sialolithiasis is the most frequent cause of obstructive and chronic inflammatory symptoms of the salivary glands, with an incidence of 1:10,000 to 1:30,000 inhabitants [1]. Sialolithiasis is most often found in the submandibular gland (60–85%), and the parotid gland is affected in about 30 to 40% of the cases [2]. The diagnosis is usually made by ultrasound [3], which is favored over the alternative imaging tools such as computed tomography (CT) scan, cone-beam CT or magnetic resonance imaging (MRI) [4]. Sialolithiasis has to be differentiated from mimics such as calcified lymph nodes, phleboliths or atherosclerotic vascular changes or calcified scars [5]. In tumorous lesions, particularly in malignant salivary gland tumors, MRI belongs to the standard imaging, as soft tissues can be assessed more accurately [4]. Although rarely, calcifications can occur within salivary gland tumors [6,7]. Diagnosis can be easily established if the tumor can be clearly circumscribed and internal calcifications do not result in difficulties concerning any differential diagnosis [8].

In the following, we would like to highlight the pitfalls in the differential diagnosis of sialolithiasis of the parotid gland, presenting two rare cases of calcified malignant tumors (adenoid cystic carcinoma (AdCC) and acinic cell carcinoma (ACC)), which were mistaken for salivary stones.

## 2. Case 1

A 37-year-old male patient presented to our department with permanent swelling and recurrent painful episodes in the left parotid gland without periprandial association or other symptoms. Apart from a testicular tumor in the past, the patient did not suffer from any other diseases. The ultrasound examination revealed a hyperechoic reflex with dorsal sound shadowing in the region of the distal part of the parotid duct matching the diagnosis of sialolithiasis (Figure 1a). No signs of a salivary duct obstruction could be detected through an ultrasound. Sialendoscopy had already been performed elsewhere in the past without detecting any signs of sialolithiasis. Additionally, a multiparametric MRI including diffusion weighting had been performed (Figure 1b) to exclude neoplasia as a differential diagnosis. On imaging, no tumorous lesion of the parotid gland could be identified. Sialendoscopy was performed in our department. As no stone was visible, extracorporeal shock wave lithotripsy (ESWL) was indicated based on the suspicion of (intraparenchymal) sialolithiasis. In the further course, altogether three sessions of ESWL were performed, all without significant therapeutic success. Due to the persistent complaints (periprandial painful swelling) and changes within the parotid gland (hyperechoic reflex) during the ultrasound examination and surgical exploration. If necessary, parotidectomy was indicated after consultation with the patient. However, the patient did not decide to present in our department for any kind of treatment at that time. Seven years later, the patient presented again to our department, and sonographic re-examination revealed the same hyperechoic reflex with dorsal sound shadowing surrounded by blurred-margined hypoechoic tissue, which was suspected to correspond to scar tissue due to an inflammatory reaction (Figure 1c). Again, no tumorous lesion in the parotid gland could be demarcated on ultrasound. Due to the unchanged findings and complaints, surgical exploration of the parotid gland resulting in partial superficial parotidectomy with the dissection of the facial nerve and application of botulinum toxin to the surrounding gland tissue was performed. Intraoperatively the buccal branch of the facial nerve was found to be embedded in markedly scarred tissue. However, it could be dissected out of the scarred area and preserved. The postoperative facial function was normal (House-Brackmann Grade I) [9]. Surprisingly, histopathological examination revealed an incomplete resected (R1) high-grade AdCC with a central calcified well-circumscribed tumor nodule and pronounce perineural sheath infiltration (Pn1). No angioinvasion was present, and no lymph node metastasis was not detected on preoperative imaging. TNM-classification was pT3 cN0 L0 V0 Pn1 R1 high grade (Figure 2). The postoperative computed tomography of the neck and thorax showed no evidence for regional or distant metastases. According to the recommendation of our interdisciplinary tumor board, revision surgery was indicated. Revision surgery consisted of radical parotidectomy with the removal of the residual parenchyma of the parotid gland and the main trunk, and all branches of the apparently infiltrated facial nerve, as well as ipsilateral neck dissection (levels Ib-V). Dynamic facial nerve reconstruction was performed using branches harvested from the cervical plexus. Planned follow-up consists of regular follow-up examinations, including clinical examination, ultrasound, MRI and CT scans.

## 3. Case 2

A 47-year-old female patient presented to our hospital with recurrent episodes of painful swelling in the left parotid region without periprandial association. The patient had already been treated with multiple antibiotic therapies due to suspected sialolithiasis in another hospital without any relief of symptoms. Her medical history was unremarkable except for chronic nicotine abuse. Our first ultrasound examination initially suggested sialolithiasis with intraparenchymal location (Figure 3a). Therefore, ESWL was planned. However, the second ultrasound examination before conducting ESWL revealed a tumorous lesion with intralesional calcifications (Figure 3b). For that reason, we initiated a multiparametric MRI examination, including diffusion and perfusion weightings, confirming the presumption of a tumor with the suspicion of a calcified hemangioma (Figure 3c). To establish a final diagnosis and to plan further management, a core needle biopsy (CNB) of the parotid lesion was carried out. The histopathologic examination of the tissue material showed an ACC. A total parotidectomy with preservation of the facial nerve and elective neck dissection (levels Ib-V) on the left side was performed without any complication. The postoperative function of the facial nerve was normal (House Brackmann Grade I). Definitive histology revealed a low-grade pT2 pN0 (0/15) L0 V0 Pn0 ACC. The follow-up clinical and imaging examinations showed no evidence of recurrence more than 10 years after treatment.

## 4. Discussion

Gland-preserving, minimally invasive interventional treatment options, such as sialendoscopy, intra- or extraductal lithotripsy, or a combined procedure, became the gold standard at large salivary gland centers in the treatment of salivary calculi of the parotid in recent years [10]. As gland resection is avoided in over 95% of the cases, histopathological examination of glands is rather unusual nowadays.

Sialolithiasis represents the most common cause of calcification within the parotid gland, while other calcifying lesions of the parotid gland are rare and mostly benign. Differential diagnoses include vascular malformations, calcified lymph nodes [5,11], and even malignant parotid gland tumors can harbor calcifications in any tumor area [6]. In addition to the typical intraductal stones, salivary calculi can also occur extraductally or intraparenchymally in up to 10–20% [2], complicating the differential diagnosis. The two cases mentioned above point out major differential diagnostic pitfalls on the ground of large intraparenchymal calcifications in the parotid gland. Both cases revealed calcifications in typical locations, pointing to sialolithiasis. Especially in the first case, there was no well-demarcated tumor recognizable on ultrasound examination, and even color Doppler sonography depicted no relevant perfusion in the hypoechoic surrounding tissue, which would have suggested a malignant tumor. In these rare cases, early histological diagnostic assurance would have been indispensable. Therefore, sonographic detection of a hyperechoic reflex with dorsal acoustic shadowing, without obstruction signs in the proximal duct, missing visualization of sialolithiasis in sialendoscopy, and missing typical complaints like periprandial symptoms should raise suspicion for a malignant affection and should be clarified at least by means of further imaging like multiparametric MRI. However, the first case shows that due to the diffuse tumor growth without defined tumor borders, even MRI could not clearly depict a tumor.

However, even in the case of excluding a tumor from imaging, a surgical exploration rather than a “blind” ESWL of the finding should be preferred. Conversely, surgical exploration of an inflamed, scarred region with a hidden malignancy bears the risk of opening the tumor, which in turn can lead to a worse prognosis by distributing tumor cells in the surgical wound. Additionally, a two-timed approach contains the difficulty of finding exactly those spots where the tumor was incompletely resected during the first session and increased the risk of facial nerve palsy due to the scarred tissue. Furthermore, the probability of a malignant tumor of the parotid gland is extremely low, as, e.g., AdCC represents only 2% and ACC about 1% of all neoplasms of the parotid gland [12,13], which is why a cost–benefit assessment based on the morbidity of the surgical approach (e.g., the potential of facial nerve palsy) should be considered carefully. The association of malignant tumors with incidental concurrent true calculi or salivary malignancies with large calcifications within the tumor mimicking salivary calculi have—to the best of our knowledge—only been described in two cases so far [7,14]. The fact that the differential diagnosis could be a centrally calcified AdCC was very surprising in the first case mentioned above, as prior imaging and clinical presentation 7 years before did not give rise to suspicion of malignancy. Since both AdCC and ACC are slowly growing tumors with indolent clinical behavior, they can easily be confused with benign entities [13]. However, due to AdCC’s potential to develop distant metastases very late in the process and their 10-year-overall-survival rate of <50% [15], an earlier diagnosis would certainly have been desirable. Fortunately, our patient had no distant metastases in computed tomography seven years after the first presentation. Therefore, in suspicious cases beyond the conventional B-Mode ultrasound, color Doppler sonography should first be conducted to discover possibly existing atypical vascularization. Secondly, a multiparametric MRI examination should be performed in ambiguous cases. In cases with large calcifications, diffusion and perfusion weightings often do not provide further information as they cannot depict calcifications due to missing signals. Therefore, additionally, special sequences, such as susceptibility weighted imaging (SWI), should be considered visualization of even small calcifications is possible. They could give further hints on the correct diagnosis in combination with other weightings. Further, MRI should be preferred over CT for differential diagnostics, as soft tissue can be better visualized and therefore tumor tissue (e.g., around the calcifications) can be better detected. If doubts remain, CNB or surgical exploration should be planned. Fine needle aspiration biopsies (FNA) are inferior to CNB because of the high diversity of salivary gland tumors, overlapping cytologic findings, and lack of tissue structure and therefore cannot be recommended as the first choice in diagnostics of parotid gland neoplasms [16,17]. If surgical exploration is planned, it should be conducted with intraoperative frozen section diagnostics for malignancy detection and to avoid a two-stage procedure.

## 5. Conclusions

These cases show that the most obvious diagnosis is not always the right one. However, intraparenchymal sialolithiasis is definitely more common than calcified carcinomas of the parotid gland, and imaging can reveal no definable tumor mass. Therefore, the therapeutic approach should be considered carefully in each individual case, and rare differential diagnoses, such as malignancies, should also be considered. A high suspicion index of the need for further diagnostics in cases with calcifications is practical and could include missing periprandial symptoms, no obstruction signs in the proximal duct, and missing evidence of sialolithiasis in sialendoscopy. Moreover, in cases where a calcified benign tumor is suspected in imaging, surgical resection should be attempted if there is a low risk for facial nerve injury. Otherwise, a close follow-up for the exclusion of another entity is advisable.

## Figures and Tables

**Figure 1 jcm-11-03329-f001:**
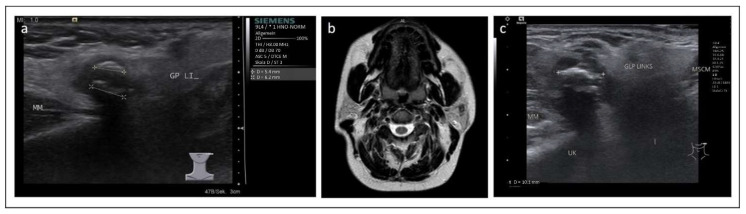
Ultrasound and MRI findings of a 37-year-old male patient with recurrent swelling and pain of the left parotid gland. (**a**) Initial ultrasonography examination revealing two echogenic reflexes (see markings) with dorsal sound cancelling in the anterior part of the left parotid gland. The remaining gland parenchyma shows hypoechoic changes, consistent with chronic sialadenitis. MM = masseter muscle, GP LI = left parotid gland. (Siemens Medical Solutions, Acuson S3000). (**b**) Initial MRI examination of T2 TSE transversal sequence showing unspecific inflammatory changes. (**c**) Pre-operative ultrasonography examination revealing the echogenic reflexes (see marking) with dorsal sound cancelling in the anterior part of the left parotid gland – but larger in comparison. MM = masseter muscle, GLP links = left parotid gland, UK = mandibule, MSCM = sternocleidomastoideus muscle. (Siemens Healthineers, Acuson Sequoia).

**Figure 2 jcm-11-03329-f002:**
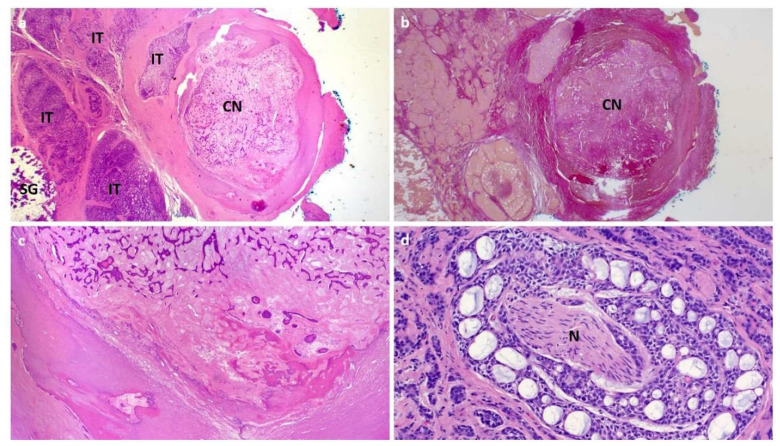
Histopathologic findings of the first case revealing a calcified adenoid cystic carcinoma of the left parotid gland. (**a**) Low power photomicrograph illustrating a 5 mm encapsulated and extensively sclerosed and calcified tumor nodule (CN) surrounded by widely invasive tumor (IT) and a rim of salivary gland tissue (SG) at the periphery. (**b**) EvG stain highlighting the sclerosed/calcified nodule. (**c**) High power showing adenoid cystic carcinoma with regressive changes within the calcified nodule (upper field) and adjacent calcification (lower field). (**d**) Higher magnification of the extensive perineural invasion (N = nerve) seen in the invasive tumor component. No remnants of a pleomorphic adenoma were detected.

**Figure 3 jcm-11-03329-f003:**
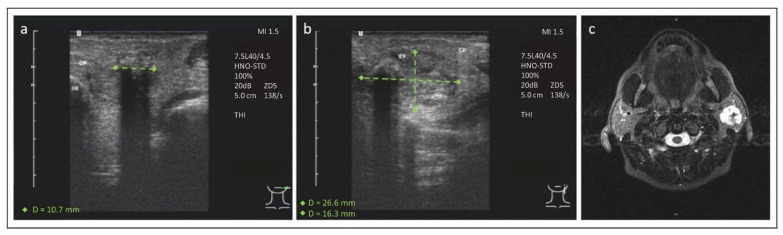
Ultrasound and MRI findings of a 47-year-old female patient with recurrent swelling and pain of the left parotid gland. (**a**) Initial ultrasound examination revealing an echogenic reflex with dorsal sound cancelling in the central part of the left parotid gland. UK = mandibule, GP = left parotid gland. (Siemens Medical Solutions, Acuson S3000). (**b**) A second ultrasound examination showing the echogenic reflex within a slightly delimitable tumor of 27 to 16 mm of the left parotid gland. RF = tumor, GP = left parotid gland (Siemens Medical Solutions, Acuson Sequoia). (**c**) MRI examination with T2 stir transversal sequence showing an inhomogeneous tumor with high signal intensity.

## Data Availability

No new data were created or analyzed in this study. Data sharing is not applicable to this article.

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
