# Peer review of "Calcification in Salivary Gland Cancer Mimicking Sialolithiasis—A Diagnostic Pitfall on Imaging: Report of Two Cases and Brief Review of the Literature"

_jcm, 2022, doi:10.3390/jcm11123329_

Round 1
Reviewer 1 Report
The conclusions are too long compared to the discussion section. They should be presented in a clearly and informative manner and they should result directly from the diagnostic considerations presented in the discussion section. All diagnostic and clinical doubts and treatment considerations regarding the differential diagnosis and the choice of diagnostic and treatment methods should be moved to the discussion section.
The manuscript does not meet the requirements of a systematic review article, but meets the criteria of case reports / letters to editor.
Please consider introducing the most recent bibliography regarding presented issues
The GNB abbreviation in row 153 should be explained. All abbreviations should have been defined the first time they appeared in the manuscript, for example MRI, AdCC (row 43 vs 73, 34 vs 53), and in following parts of the manuscript only previously defined abbreviations shoud be used, unify the abbreviation policy in the whole manuscript.
The size of letters, the reference style, the methods of the bibliography presenting and citation should be unified in the whole manuscript and adapted to the requirements and recommendations of the journal. Why the bigger size of letters in selected parts of the discussion section was used?
Author Response
Point 1: The conclusions are too long compared to the discussion section. They should be presented in a clearly and informative manner and they should result directly from the diagnostic considerations presented in the discussion section. All diagnostic and clinical doubts and treatment considerations regarding the differential diagnosis and the choice of diagnostic and treatment methods should be moved to the discussion section.
Response 1: Thank you for this advice. We have shortened the conclusions section to the essential statements.
Point 2: The manuscript does not meet the requirements of a systematic review article, but meets the criteria of case reports / letters to editor.
Response 2: Thank you for this comment. We changed the title of the manuscript to „Calcification in salivary gland cancer mimicking sialolithiasis – a Diagnostic Pitfall on imaging: Report of two cases and brief review of the literature“.
Point 3: Please consider introducing the most recent bibliography regarding presented issues.
Response 3: Thank you for this comment. We again checked the bibliography. For the presented issues the most relevant literature sources where chosen. Due to the rare topic, larger studies concerning certain questions are not always available from recent years.
Point 4: The GNB abbreviation in row 153 should be explained. All abbreviations should have been defined the first time they appeared in the manuscript, for example MRI, AdCC (row 43 vs 73, 34 vs 53), and in following parts of the manuscript only previously defined abbreviations shoud be used, unify the abbreviation policy in the whole manuscript.
Response 4: Thank you for this comment. We apologize for this error and have added the explanation (core needle biopsy) and changed the abbreviation itself accordingly to CNB. We further unified the abbreviation policy in the manuscript.
Point 5: The size of letters, the reference style, the methods of the bibliography presenting and citation should be unified in the whole manuscript and adapted to the requirements and recommendations of the journal. Why the bigger size of letters in selected parts of the discussion section was used?
Response 5: In the original word document the size and font of letters was consistent. Therefore, the differences in size/font is probably due to the conversion to PDF after submitting the manuscript? Of course, this should be corrected before the manuscript is published. Thank you very much for this advice. We further again proved the reference style of the whole manuscript and unified the distance between text and bibliography. Thank you for this comment.
Reviewer 2 Report
This case report highlighted the pitfall in interpreting the ultrasound findings of salivary gland lesion.
When dealing with any head and neck conditions, calcification should always raise suspicion of a malignancy and any clinicians must be take extra precaution in managing them. There is a role of fine needle aspiration when there is ambiguity in obtaining the diagnosis.
Though, the manuscript is well written, the reporting of these 2 cases does not address this issue.
Author Response
Point 1: This case report highlighted the pitfall in interpreting the ultrasound findings of salivary gland lesion.
When dealing with any head and neck conditions, calcification should always raise suspicion of a malignancy and any clinicians must be take extra precaution in managing them. There is a role of fine needle aspiration when there is ambiguity in obtaining the diagnosis.
Though, the manuscript is well written, the reporting of these 2 cases does not address this issue.
Response 1: Thank you very much for your comment. In fact fine needle biopsy is not addressed in this issue. This is due to the fact, that fine needle biopsy in the diagnosis of parotideal space lesions is highly controversial and also highly dependent on the quality and experience of the examiner as well as the type of neoplasm. The diagnosis of fine needle biopsy is particularly difficult due to the rarity of malignant salivary gland neoplasms and the diversity of carcinoma types with overlapping cytologic findings of benign tumors and low-grade carcinomas. Therefore, in our hospital, a large salivary gland center, we do not perform fine-needle biopsies but core-needle biopsies, as they provide higher diagnostic accuracy due to the larger amount of tumor material and as the tissue structure is obtained. This is also taken into account in the manuscript in lines 153 and 160. (Examples of relevant literature: Hanege et al. 2020, Jering et al. 2022, Stanek et al. 2019, Zbären et al. 2018, Iftikhar et al. 2020).
To accommodate your request, we added a sentence on the role of fine needle biopsy in parotid gland lesions in the discussion section.
Reviewer 3 Report
Calcification of salivary gland tumors is not so strange. RNM images show a tumor in both cases and should have been diagnosed by an expert radiologist. There are other methods as perfussion-difusion MR that could have been employed for an accurate diagnosis. I think that thes two cases are not relevant and the way the authors managed the patients are not adequate. In the first case, for example, they perform a neck dissection in a adenoid cystic carcinoma, which is a tumor that usually don´t cause lymph node metastasis.
Author Response
Point 1: Calcification of salivary gland tumors is not so strange.
Response 1: Calcifications occur in malignant salivary gland tumors. However, these are usually microcalcifications. On the other hand, the occurrence of macrocalcifications in the typical localization of the Stenson’s duct are rare. In our salivary gland center, in which about 280 patients with parotid gland tumors undergo surgical treatment every year, only these 2 cases have been known since 2000. Therefore, in our opinion, the two cases are relevant and worth mentioning.
Point 2: RNM images show a tumor in both cases and should have been diagnosed by an expert radiologist. There are other methods as perfussion-difusion MR that could have been employed for an accurate diagnosis.
Response 2: Thank you for this comment. In fact, MRI was diagnosed by expert radiologists with many years of experience in the diagnostic field of salivary glands before initiating the respective therapy. Of course, all weightings relevant at the time the diagnostics were carried out were performed and assessed.
Point 3: I think that thes two cases are not relevant and the way the authors managed the patients are not adequate. In the first case, for example, they perform a neck dissection in a adenoid cystic carcinoma, which is a tumor that usually don´t cause lymph node metastasis.
Response 3: Thank you for this comment. Certainly, adenoid cystic carcinomas are better known for distant metastasis. But especially in larger tumors (T3, T4) and high-grade transformation lymph node metastases are not uncommon (e.g.: International head and neck scientific group 2017, Min et al. 2012, Megwalu et al. 2017, Hellquist et al. 2016). In the first case mentioned the patient hat a pT3 high grade tumor. Therefore, a radical surgical treatment with neck dissection is indicated.
Round 2
Reviewer 2 Report
Concerns have been addressed.
Author Response
We are pleased to have addressed your concerns satisfactorily.
Reviewer 3 Report
It is not a diagnostic pitfall. It is a wrong interpretation of images. Diffusion-perfussion MRI is mandatory in these cases.
Author Response
Point 1:
It is not a diagnostic pitfall. It is a wrong interpretation of images. Diffusion-perfussion MRI is mandatory in these cases.
Response 1:
Thank you for again for your review. As you of course quite correctly point out, the investigation of salivary gland tumors requires a multiparametric MRI scan for proper diagnosis. Both patients received a mutliparametric MRI examination including diffusion imaging. Indeed, perfusion imaging was not performed in the first case (2012) but in the second case. However, due to the extended calcifications in the first case (tumor completely calcified), the sequences would not have been of any additional use, as they are without signal in the area of calcifications. More helpful would be the susceptibility-weighted sequences (SWI), which can visualize even the smallest calcifications.
To meet your concern, we have considered your comment in the discussion and added that a multiparametric MRI examination with diffusion and perfusion weightings are essential in the diagnosis of salivary gland tumors as they can provide conclusions about the tumor entity. In the case of macrocalcifications, however, the additional performance of SWI should be considered for visualization of calcifications, which is not possible sufficiently with the above-mentioned sequences.
We hope that we could address your concerns satisfactorily.